# The Effect of Microwave Irradiation on the Representation and Growth of Moulds in Nuts and Almonds

**DOI:** 10.3390/foods11020221

**Published:** 2022-01-14

**Authors:** Eva Popelářová, Eva Vlková, Roman Švejstil, Lenka Kouřimská

**Affiliations:** Department of Microbiology, Nutrition and Dietetics, Faculty of Agrobiology, Food and Natural Resources, Czech University of Life Sciences Prague, 140 00 Praha, Czech Republic; popelarova@af.czu.cz (E.P.); vlkova@af.czu.cz (E.V.); svejstil@af.czu.cz (R.Š.)

**Keywords:** microwave treatment, moulds, peanuts, pistachios, almonds

## Abstract

Microwave (MW) irradiation is a non-destructive method that can be applied as an alternative method to inhibit the growth of microorganisms. The present study evaluated the effect of MW irradiation on the occurrence of moulds in nuts and almonds. Samples of unshelled natural almonds, pistachios, and in-shell peanuts were treated with different doses of MW irradiation (2400–4000 W). The effect of MW irradiation on mould counts was evaluated by cultivating immediately after irradiation and after 3 and 6 months of storage. The most represented genera in all analysed samples were *Aspergillus* (68%), *Penicillium* (21%), and a small amount *of Cladosporium* (3%). Mould numbers significantly decreased after MW treatment. The treatments with MW irradiations at 3000 and 4000 W significantly reduced the mould colony counts, and their effect persisted during storage; irradiation at 2400 W was partially effective. The strongest effect of MW irradiation was observed in in-shell peanuts. MW irradiation seems to be a promising method for maintaining the microbiological quality of nuts.

## 1. Introduction

Nuts, dry fruits, and small seed plants are very popular ingredients in the human diet because of their high content of valuable proteins, essential polyunsaturated fatty acids, fat-soluble vitamins (A, D, E, and K), water-soluble vitamins (B1, B6, and C), minerals, and trace elements (Ca, S, B, Mg, P, Zn, Fe, and Mn) [1]. Compared to other foods, nuts contain relatively small amounts of carbohydrates and therefore have a low glycaemic index. A nut-fortified diet can lower cholesterol and blood pressure [2].

The relatively high percentage of fat contained in nuts also carries a potential risk of rancidity. The presence of water, improper handling, and storage conditions increase the risk of spoilage of nuts by the activities of microorganisms in the presence of oxygen. Higher moisture storage may cause mould development, especially under nutshells. Contamination of nuts by aflatoxin-producing fungi is a major problem affecting the production, preservation, and sale of nuts [3], and leads to economic restrictions for the producing regions [4]. Mycotoxins are a major global problem because mould-infected products can cause serious illnesses, which limit the worldwide consumption of nuts.

Restriction or elimination of undesirable microorganisms can be achieved using conservation techniques. Conservation techniques attempt to limit the reproduction process of the microorganism in nuts predominantly by removing oxygen, reducing the water content, or combining these or several other methods. Microwave (MW) irradiation is one these treatments. It is used in several industries, such as wood, papermaking, textiles, ceramics, and food [5,6]. It is most commonly used for heating (defrosting, boiling, baking, roasting, and drying [7]), as well as for bleaching (inactivation of enzymes), pasteurisation, and sterilisation [8]. It quickly reduces moisture content and contributes to the inhibition of enzymes involved in lipid oxidation, which not only preserves sensory properties, but also maintains quality during storage [9]. Generally, microwave drying affects the lipid, protein, and other components, and could have a positive effect on drying speed, elasticity, colour, taste, nutritional value, microbial stability, rehydration capacity, and crispness of treated products [10].

MW irradiation is a non-destructive method where the absorbed energy of MWs increases the temperature of food due to friction between individual water molecules [11] and causes the inactivation of microorganisms by heat [12]. MW treatment may disrupt cell walls of microorganisms and cause thermal denaturation of their proteins, but the mechanisms of bacterial inactivation by heat and the factors affecting this process are still not fully understood [13]. The thermal effect is the basic contribution of MW irradiation to the destruction of microorganisms; however, some studies [14,15] have also indicated the non-thermal effect of MWs. However, according to Dudley et al. [16], there are no known molecular resonance processes that occur at the typical microwave oven frequency of 2.45 GHz, and any proposed mechanisms for non-thermal microwave effects that involve or require such a process are unlikely. The final effect depends on the water content and physical properties of the treated raw materials [17]. MWs may be an alternative to chemical methods because their application does not leave any undesirable residues [18].

The main advantages of MW irradiation are increased energy savings, shorter processing times, and lower operational costs [10]. Further, MW treatment is advantageous because there are minimal changes in organoleptic properties; therefore, it can be an alternative to other hygienic techniques for the conservation of dried food products [7,19]. The major disadvantage of MW heating is the unequal distribution of temperature, which creates cold and warm zones. This could cause incomplete deactivation of microorganisms [20]. Thus, better construction of MW systems is important for using the MW process as a reliable pasteurisation method [17]. Guo et al. [10] stated that bacterial cells can reach higher temperatures than the surrounding liquid because of selective MW heating, which leads to the rapid destruction of microorganisms.

Contamination of stored grain with insects, insect fragments, fungi, and mycotoxins is a major concern in the grain industry. MW irradiation destroys bacteria such as *Escherichia coli*, *Enterococcus faecalis*, *Staphylococcus aureus*, and *Salmonella* spp. The use of MW technology for the disinfection of cereals and legumes was reported by Yadav et al. [18]. The effect of MW irradiation on microorganisms has also been documented in milk. The number of fungi and yeasts decreased from 600–0 CFU/mL when 1500 W was applied for 30 s [21]. Dańczuk et Lomotowski [22] demonstrated high efficiency in the destruction of pathogenic microorganisms in soil, liquid wastes, and insects in grains. Guo et al. [10] investigated the inhibitory effects of MW irradiation on the growth of microorganisms and confirmed the reduction in bacterial counts [23]. Zhang et al. [24] suggested that MW irradiation can significantly increase the preservation of the quality of wet wheat and rice-based noodles by reducing the number of bacteria, yeast, and fungi, but only with the addition of 0.20% citric acid and 150 ppm vitamin C. The effect of MW irradiation on *E. coli* was demonstrated by Hamoud-Agha et al. [20]. Pucciarelli et al. [25] confirmed the inactivation of the genus *Salmonella* using MWs. Bauza-Kaszewska et al. [26] studied the same bacteria and found that inactivation depends on the power of the emitter. Daňczuk et Lomotowski [22] used MW energy to eliminate undesirable bacteria from sewage sludge. The efficacy of MW treatment has been proven; the most sensitive were *Salmonella* and *E. coli* and the highly resistant were thermophilic bacteria and *Clostridium*. Lakins et al. [27] and Reddy et al. [28] examined the use of MW irradiation for mould inactivation.

MW irradiation is more often used to suppress the growth of bacteria, but the action of MWs on moulds, especially in connection/context with the treatment of nuts, has not been widely reported. Therefore, this study investigated the effect of different MW irradiation intensities on the suppression of growth of moulds in nuts and almonds immediately after their treatment, as well as after the storage of samples for 3 and 6 months.

## 2. Materials and Methods

### 2.1. Analysed Samples

Samples of unshelled natural almonds (*Prunus dulcis*), in-shell pistachios (*Pistacia vera*), and natural in-shell peanuts (*Arachis hypogaea*) were supplied by the IBK Trade Company (Czech Republic). The samples were intended for sale on the market and were therefore not artificially spiked by moulds.

The experiment focused on the effect of MW irradiation on mould counts in nuts and almonds immediately after irradiation and after 3 and 6 months of storage. The tested samples were stored under conditions simulating their storage in food stores (in poly-ethylene bags, in the dark, with access to air and at a constant temperature of 22 °C).

### 2.2. Treatment Conditions

MW treatment was performed on an experimental continuous line (ROmiLL, Brno, Czech Republic). The samples were placed on a conveyor belt (belt speed 1 cm/s; the thickness of the layer corresponded to the size of one sample of walnut or almond) and passed through the exposure tunnel for 50 s, where they were irradiated (different wattages of MW irradiation, from 2400–4000 W, were applied at a frequency of 2450 MHz). The power of the microwave emitter was set according to the nature of the sample and the requirements for the final product. In case of natural almonds, both treatments that do not change sensory properties (2400 W) and treatments that obtain a lightly roasted product (3000 and 4000 W) were tested. As natural peanuts have a strong beany flavour, the goal was to roast them during the treatment (4000 W applied). In the case of in-shell pistachios, the aim was to stabilize them while maintaining the original sensory properties (3000 W). The treatment regimens were chosen based on a preliminary test.

The MW treatment was followed by a heating zone (190 cm long) supplied by hot air at 84–86 °C. The aim of this tempering tunnel was to achieve an even temperature distribution in the treated product, because when the samples were treated only with microwaves, the samples were unevenly heated and locally overheated. The final maximum temperature of the samples varied between 100 to 120 °C depending on the treatment. The control group of all the samples without MW treatment was also monitored.

### 2.3. Microbial Analysis

The horizontal ISO 21527–2:2008 method for the enumeration of yeasts and moulds was applied using Dichloran-Glycerol (DG18) Agar Base (Oxoid CM0729) (Oxoid Ltd., Basingstoke, UK) for mould cultivation [29]. Briefly, 10 g of each sample was aseptically transferred to an Erlenmeyer flask containing 90 mL of 0.1% peptone water, shaken on a platform shaker for 7 min, and aseptically diluted, using ten-fold dilution, up to 10^−2^ dilution. Next, 0.2 mL of each dilution was immediately transferred using the spread-plate method onto culture plates containing DG18. The culture plates were incubated for 5 days in a thermostat (laboratory thermobox LBT 168, ATS, Czech Republic) at 25 °C. Each determination was performed in triplicates. Colony-forming units (CFUs) were counted after cultivation. Fibrous microorganisms (moulds) were determined as flat or fluffy colonies, often coloured.

The genus representation was also tentatively investigated. The colonies were first categorised according to their macroscopic characteristics (colour, shape, and structure of the colony) and then on the microscopic features (morphology, fibre structure, and fructification organs) according to Kirk et al. [30].

### 2.4. Statistical Analysis

The results were evaluated using Statgraphics Centurion XV 15.2.05/2007 (StatPoint Technologies, Inc., Warrenton, VA, USA) and ANOVA multiple range test with Scheffé’s post-hoc test at a significance level of α = 0.05.

## 3. Results

The average mould counts (log CFU/g sample) in the samples immediately after MW treatment and after 3 and 6 months are shown in Table 1. The mould counts ranged from <1 log CFU/g (detection limit of the method) to 2.83 log CFU/g. The comparison was performed between different magnitudes of MW energy treatment within the same type of sample and at the same time point. In all cases, there was a significant decrease in the mould counts in samples after MW treatment compared with the control without irradiation.

MW irradiation had a reasonable decreasing effect on the mould counts before storage depending on the performance dose. In the case of almond samples, irradiation at 2400 W did not have a significant effect, while irradiation at 3000 and 4000 W caused a significant decrease in the mould counts. Regarding peanut samples, MW irradiation significantly decreased the mould counts below the detection limit of 1 log CFU per gram of the sample.

The counts of mould colonies also reduced significantly in pistachios after MW treatment. Generally, MW irradiation at 3000 W and higher had a significant effect on the reduction in the number of moulds in all samples immediately after treatment.

The effect of MW treatment persisted throughout the storage period of 6 months. The reductions in the mould counts compared with non-treated samples remained significant throughout the experiment.

Tentative genus identification of the colonies detected in the samples showed that mostly *Aspergillus* spp. and *Penicillium* spp. were present in the samples. Colonies of *Cladosporium* spp. were also found in almond and peanut samples. All the genera were susceptible to MW irradiation since their counts decreased after MW irradiation. *Aspergillus* was found in all the samples (Figure 1).

## 4. Discussion

To our best knowledge, the number of experiments focused on investigating the effect of microwaves on moulds in nuts and almonds is very limited. In particular, data are available showing the effect of microwave irradiation on bacteria. The major mechanism for the inactivation of microorganisms using MW (including the moulds) is heat [31,32]. MW irradiation generates a force on water molecules that leads to their oscillation, which causes the material to heat up [16,33]. Thus, the main goal of our study was to investigate the effect of MW irradiation on the suppression of growth of moulds in real samples of nuts and almonds. The effect of irradiation was also evaluated during the storage of the samples.

According to Guo et al. [10], MW irradiation can be used to limit the potential microorganisms in food. Inactivation of pathogens and reduction in the number of microbial colonies in food during MW sterilisation has been confirmed. In addition, other authors [14,34] have also reported the ability of the MW to eliminate counts of microorganisms, which supports our findings regarding the effect of MW treatment on the reduction in the fungal count. Our results showed that there was a significant decrease in the mould counts in all the samples treated with MW irradiation. With increasing irradiation intensity, a more significant reduction in the number of molds was achieved compared with the untreated control. The strongest effect was observed in irradiation of in-shell peanuts and pistachios with 4000 W, where the number of colonies decreased by more than one order. These findings are consistent with those of Reddy et al. [28], who proved that the destruction of pathogens increases with the intensity of MW energy. They investigated the relationships between microwave operating conditions, degree of inactivation of *Fusarium graminearum* in wheat, and the resulting seed quality. Their results showed that eradication of the pathogen increased with the total MW energy irradiated. An investigation into the effect of MW on the fungal pathogens of winter wheat was also conducted by Knox et al. [35]. MW significantly reduced the levels of seed contamination and was particularly effective at controlling the growth of *Fusarium* spp.

The efficacy of MW in the disinfection of in-shelled Brazil nuts contaminated with aflatoxin-producing fungal strains was investigated by da Silva et al. [4]. No significant reduction in the number of fungi in the outside shell was observed, but there was a reduction in colonisation of the kernel and inside shell after treatment with MWs. On the contrary, in the present study, the counts of moulds present on the outer shell of peanuts decreased below the detection limit (1 log CFU/g) after MW irradiation at 4000 W.

Cavalcante et Muchovej [36] investigated MW irradiation as an alternative method of treatment of seeds (soybean, peanut, bean, wheat, or corn) for the control of seed-borne fungi, and found that MW irradiation of seeds appears to be a promising method for eliminating fungal spores without using fungicides. Similarly, in our study, the percentage of fungi after MW treatment generally decreased compared with that of the control. After irradiation, the counts of *Aspergillus* spp., *Penicillium* spp., and *Cladosporium* spp. in tested samples decreased to less than half of the original counts when using 3000 or 4000 W of MW irradiation. The counts of *Penicillium* spp. decreased to half of the original counts after using the previously mentioned irradiation. Chen [37] suggested that MW irradiation can be a practical tool for textiles contaminated with *Aspergillus niger*. This study shows that *Penicillium* was the most reduced genus of fungi, although the *Aspergillus* spp. were also reduced by MW treatment.

Since the level of primary contamination with moulds plays a crucial role and for our experiment where used samples were not visibly infected, the decrease after irradiation was moderate compared to other studies when spiked samples were used [38]. Dababneh [39] proposed a MW process for the microbial decontamination of spices and herbs from local markets. Sixteen tested samples were artificially contaminated with aerobic mesophilic bacteria and moulds. The treatment of spices and herbs with MW oven resulted in a reduction in the microbial population, and the total mould counts decreased by 1–3 log. In our experiment, maximum reduction 2 log was achieved. Basaran et Akhan [3] carried out experiment with hazelnuts spiked with aflatoxigenic *Aspergillus parasiticus* at a concentration of 10^6^–10^7^ CFU/g and showed a reduction of nearly 3 log after treatment with MW at 1250 W for 120 s. In our experiment, we managed to reduce the level of *Aspergillus* spp. by more than one order using MW irradiation at 2400 W and by more than two orders using MW irradiations of 3000 and 4000 W, respectively. This reduction was not so significant, because in contrast to the study of Basaran et Akhan [3], we used naturally contaminated nuts with much lower initial mould concentration.

Maintaining food without moulds during long-term storage can also be a strategic issue. Lakins et al. [27] studied the growth of moulds on long-term stored bread for military purposes. The use of MW irradiation confirmed the extension of the shelf-life of white enriched bread for 2 months with the inhibition of mould growth and without a negative effect on quality. This is in line with our findings, where there was no increase in the number of moulds after 3 months of storage. A slight increase in the mould counts was observed only in the case of natural almonds in irradiated samples stored for 6 months. Even after 6 months, the samples treated with MW at more than 3000 W had significantly lower mould counts than the non-treated control samples. We suppose that the long-term effect depends on the storage conditions, since the occurrence of moulds is effectively reduced only during the MW treatment.

## 5. Conclusions

According to our results, the use of MW irradiation significantly decreased the counts of mould colonies in tested samples of nuts and almonds. The results indicate that moulds are reduced to a greater extent with increasing intensity of MW irradiation. In general, MW irradiation at 3000 W was sufficient for the reduction of moulds in both almonds and pistachios, and its effect persisted during the 6-month storage. MW irradiation appears to be a promising method for maintaining the microbiological quality of nuts and their shelf-life, but specific conditions for treatment, depending on the processing line and the composition of the processed material, need to be explored and set up.

## Figures and Tables

**Figure 1 foods-11-00221-f001:**
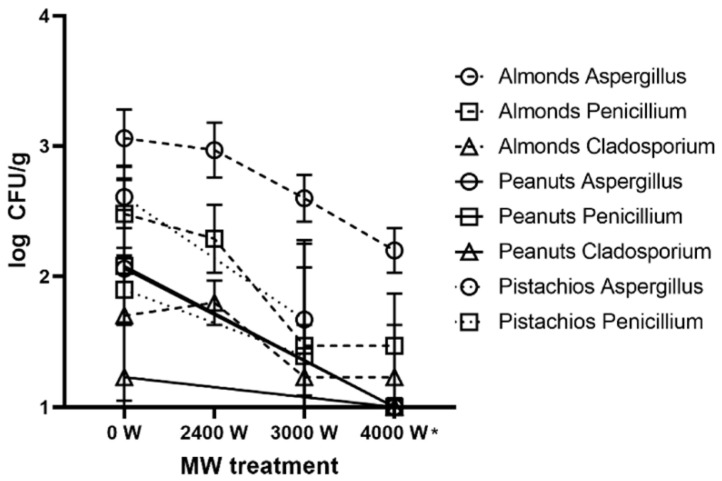
Effect of MW treatment on the mould genera representation and their counts (in log CFU/g, * significantly different values at *p* < 0.05).

**Table 1 foods-11-00221-t001:** Effect of MW treatment on mould counts during the storage of almonds and nuts.

Sample	Treatment Performance (W)	Mould Count (log CFU/g)		
		After Treatment	After 3 Months	After 6 Months
almonds	0	2.70 ± 0.19 ^b^	2.98 ± 0.12 ^d^	2.74 ± 0.05 ^b^
almonds	2400	2.60 ± 0.18 ^b^	2.70 ± 0.06 ^c^	2.78 ± 0.09 ^b^
almonds	3000	2.18 ± 0.11 ^a^	2.18 ± 0.13 ^b^	2.48 ± 0.14 ^a^
almonds	4000	2.08 ± 0.12 ^a^	1.70 ± 0.25 ^a^	2.40 ± 0.18 ^a^
peanuts	0	2.08 ± 0.09 ^b^	2.18 ± 0.12 ^b^	2.10 ± 0.14 ^b^
peanuts	4000	ND	ND ^a^	ND
pistachios	0	2.83 ± 0.11 ^b^	2.00 ± 0.05 ^b^	2.18 ± 0.10 ^b^
pistachios	3000	1.88 ± 0.16 ^a^	1.70 ± 0.25 ^a^	1.60 ± 0.08 ^a^

Notes: The results are expressed as arithmetical means ± standard deviations. ND = not detected by the method (˂1 log CFU/g). Values in the same column within one group of nuts with no common superscripts differ significantly (*p* < 0.05).

## Data Availability

Data are available on request from the corresponding author.

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
