# Peer review of "The Effect of Microwave Irradiation on the Representation and Growth of Moulds in Nuts and Almonds"

_foods, 2022, doi:10.3390/foods11020221_

Round 1

Reviewer 1 Report

The article provides interesting and practically usable information on the issue of reducing nut contamination. The article is written very carefully, clearly and with a high-quality discussion.

Recommendations / comments:

It would be useful to explain in the manuscript why the 2400 and 3000 W treatments for peanuts and the 4000 W treatments for the pistachios were not tested.

In Figure 1, it would be useful to include error bars.

There is different order of authors in the article title and citation at the first page of the article.

Reviewer 2 Report

There are nothing new or novelty in this manuscript, and appear lots of questions after review this manuscript.

1.Microwave had used for nuts preservation before, lots of paper mentioned about the antifungial properties and antibacterial activity.

2.The mechanism of anti-moulds in this manuscript was not present.

3.After microeave treatment, the lipid, protein, and other components wil be changes, but author did not mention about it.

4.The power (w) of microwave treatment for almonds was 0, 2400, 3000, 4000, but for peanuts was 0 and 4000; pistachios was 0 and 3000, the reson for design this method?

5.How about the origin status of mould?

6.How to calculate the number of mould, just by morphology via microscope?  

Reviewer 3 Report

This study was to investigate the effect of microwave irradiation on representation and 2 growth of moulds in nuts and almonds. However, MW was not the only treatment. 190s hot air was applied. There should be an comparison experiment without hot air. And effect of different microwave power was studied and it was presented that heat was the major mechanism. So the sample temperature after MW process should be illustrated. The discussion part should be shorten and restructured.

  1. Line 105-106: What is the sample temperature after 50s microwave processing? especially for different MW power?
  2. There should be a comparison experiment using the same parameter while without hot air.
  3. What is the sample thickness on the conveyor belt?
  4. Line176-181:What is this discussion for?The logicality is not proper. major mechanism is heat, low moisture may generate less heat due to dielectric properties, and at last MW irradiation generates force on water molecules which causes heat; So what do you want to express?
  5. Line 205-212 What do these previous reports support for?
  6. During the discussion part please follow this type of logic: what did you do in this study? what is the results? What did the results imply? And then any previous report to support the results or conclusion?

7.Line 220: What is the long-term effect of microwave irradiation? non-thermal effect??

  1. Line 234-235: besides MW power? any difference in sample temperature difference between your and the previous study? Theoretically Initial level could not affect the processing treatments results. Any reference to support it?

Round 2

Reviewer 2 Report

I don't think the quality and quanity of this manuscript was enough to publish in this high quality journal. 

1. This manuscript just have only one figure and table, appear low quantity on research.
2. The mechanism of microwave on antibacterial was not clear, and apply to food is uncomfortable.
3. After microwave treatment, there were some important components be changed, for example lipid oxidation, but author did not mention about it.

Author Response

Dear Reviewer,

We would like to thank you for evaluation of our manuscript and for your valuable comments and recommendations. We really appreciate the time you spent on our manuscript to make it better.

Yours sincerely,

Lenka Kourimska

Comments and Suggestions for Authors

I don't think the quality and quantity of this manuscript was enough to publish in this high-quality journal. 

  1. This manuscript just have only one figure and table, appear low quantity on research.

We monitored three different types of samples, differently treated, and stored for 6 months. Since we wanted to bring all the results together, we summarized them in one table and one graph. It is up to the editor to decide whether the manuscript is sufficient for publication in Foods. But other opponents did not have a problem with this.

  1. The mechanism of microwave on antibacterial was not clear, and apply to food is uncomfortable.

The main effect of the absorption of microwaves by most materials is heating (see lines 52-57). The brief information about non-thermal effect with the appropriate citation was added to lines 59-62. The application of microwave treatment to food is quite widespread both in households and in industrial use.

  1. After microwave treatment, there were some important components be changed, for example lipid oxidation, but author did not mention about it.

Changes of lipid, protein and other components are mentioned on lines 48-51. You are right that microwave treatment affects the oxidative stability of lipids. For example, we found that samples treated with microwaves have a higher peroxide value. However, this manuscript focuses on the effect of microwave treatment on mold growth, so we did not provide information on the effect on lipid stability to oxidation, as well as possible changes in nutritional and sensory quality.

Reviewer 3 Report

The objetive of this study was to explore the effect of MW 'radiation' . Experiments with different power setting were conducted. However, there is no designed experiments to distinguish the thermal and non-thermal (radiaiton) effect of MW. Were the results are due to the high temperure heated by MW or by the high EM field intensity? This was unclear. Add discussion for the mechnisms, i.e. the themal and non-thermal effect. 

Author Response

Dear Reviewer,

We would like to thank you for evaluation of our manuscript and for your valuable comments and recommendations. We really appreciate the time you spent on our manuscript to make it better.

Yours sincerely,

Lenka Kourimska

Comments and Suggestions for Authors

The objective of this study was to explore the effect of MW 'radiation'. Experiments with different power setting were conducted. However, there is no designed experiments to distinguish the thermal and non-thermal (radiation) effect of MW. Were the results due to the high temperature heated by MW or by the high EM field intensity? This was unclear. Add discussion for the mechanisms, i.e., the thermal and non-thermal effect.

The main effect of the absorption of microwaves by most materials is heating. Non-thermal microwave effects are effects that are not due to the increase of thermal energy of the material. According to Stuerga and Gaillard (1996a, 1996b) non-thermal effects in liquids are almost certainly non-existent, as the time for energy redistribution between molecules in a liquid is much less than the period of a microwave oscillation. Dudley et al. (2015) discussed the non-thermal microwave effect (a resonance process) in relation to selective heating by Debye relaxation processes. According to them commercial microwave ovens operate at a fixed frequency of 2.45 GHz (0.082 cm−1), which is significantly below (by many orders of magnitude) the energy required for exciting electronic or vibrational transitions. As such, there is no mechanism by which radiation can be absorbed to create an excited state that might lead to the activation of a chemical bond. They also stated that the segregation of microwave effects into “thermal” and “non-thermal” processes is only somewhat meaningful in evaluating hypotheses of microwave effects, which had led to some confusion and debate over these classifications. In their opinion, a more significant distinction is between resonant and relaxation processes, of which only the latter is likely for solutions at typical microwave frequencies. They said that there are no known molecular resonance processes that occur at the typical microwave oven frequency of 2.45 GHz, and any proposed mechanisms for non-thermal microwave effects that involve or require such a process are unlikely. For an independent molecule in the condensed phase, the current understanding is that microwave absorption can only result in a relaxation process that generates heat.

Based on this information we assume, that the non-thermal effect of MW is negligible comparing to the thermal effect.

The brief information about non-thermal effect with the appropriate citation was added to lines 59-62.

Stuerga, D.; Gaillard, P. 1996a. Microwave athermal effects in chemistry: A myth's autopsy. 1. Historical background and fundamentals of wave-matter interaction. Journal of Microwave Power and Electromagnetic Energy, 31(2), 87-100.

Stuerga, D.; Gaillard, P. 1996b. Microwave athermal effects in chemistry: A myth's autopsy. 2. Orienting effects and thermodynamic consequences of electric field. Journal of Microwave Power and Electromagnetic Energy, 31(2), 101-113.

Dudley, G. B.; Richert, R.; Stiegman, A. E. 2015. On the existence of and mechanism for microwave-specific reaction rate enhancement. Chemical Science. 6(4), 2144-2152. DOI 10.1039/c4sc03372h